# On the Origin of Raman Activity in Anatase TiO_2_ (Nano)Materials: An *Ab Initio* Investigation of Surface and Size Effects

**DOI:** 10.3390/nano13121856

**Published:** 2023-06-14

**Authors:** Beata Taudul, Frederik Tielens, Monica Calatayud

**Affiliations:** 1Laboratoire de Chimie Théorique, LCT, Sorbonne Université, CNRS, 4 Place Jussieu, F-75005 Paris, France; beata.taudul@gmail.com; 2General Chemistry (ALGC)—Materials Modelling Group, Vrije Universiteit Brussel, Pleinlaan 2, 1050 Brussel, Belgium; frederik.tielens@vub.be

**Keywords:** dft, Raman spectra, titania nanomaterials, anatase surfaces

## Abstract

Titania-based materials are abundant in technological applications, as well as everyday products; however, many of its structure–property relationships are still unclear. In particular, its surface reactivity on the nanoscale has important consequences for fields such as nanotoxicity or (photo)catalysis. Raman spectroscopy has been used to characterize titania-based (nano)material surfaces, mainly based on empirical peak assignments. In the present work, we address the structural features responsible for the Raman spectra of pure, stoichiometric TiO_2_ materials from a theoretical characterization. We determine a computational protocol to obtain accurate Raman response in a series of anatase TiO_2_ models, namely, the bulk and three low-index terminations by periodic *ab initio* approaches. The origin of the Raman peaks is thoroughly analyzed and the structure–Raman mapping is performed to account for structural distortions, laser and temperature effects, surface orientation, and size. We address the appropriateness of previous experimental use of Raman to quantify the presence of distinct TiO_2_ terminations, and provide guidelines to exploit the Raman spectrum based on accurate rooted calculations that could be used to characterize a variety of titania systems (e.g., single crystals, commercial catalysts, thin layered materials, facetted nanoparticles, etc.).

## 1. Introduction

Nanosized titania is a versatile component of many technological devices [1,2]. It can be found in (photo)catalysts for the water photosplitting reaction, in heterogeneous catalysis compositions as active phase or as support, and in self-cleaning paintings [3,4,5,6,7]. It is also present in everyday products such as paintings and makeup, and as an excipient in drugs and food additives [8,9,10,11,12]. However, it has been pointed out as probably carcinogenic, and was recently banned as an additive in the food industry in Europe in its nanosized form [13]. Indeed, most of the applications and hazards of TiO_2_, and of nanomaterials in general, are related to their surfaces, as they are extremely reactive and in constant contact with the environment. A detailed characterization of their structure is therefore needed to pinpoint the structural features responsible for the properties they exhibit.

Whereas common characterization techniques based on microscopy or diffraction cannot be applied to nanomaterials due to their small size, spectroscopic techniques are well suited as they are sensitive to both bulk and surface. They are fast, relatively inexpensive, and adapted to a wide variety of compositions and structures, from inorganic crystals to amorphous polymers and microbiological samples. In particular, Raman spectroscopy has proven to be able to unravel structural features in nanosized materials. To be properly exploited, a set of reference materials with controlled composition, structure, and conditions is desired [14]. Nevertheless, experimental Raman spectra are extremely dependent on uncontrolled parameters such as undefined composition, unwanted interaction of the sample with radiation or heat (fluorescence, resonance), equipment setup or operator’s protocol, such that different conditions may result in different spectra for the same sample. Modelling is a powerful tool to establish the relationship between well-controlled model systems and their Raman spectra, in particular allowing the identification of the origin of the vibrational Raman signals. *Ab initio* approaches appear as an experiment-free way of obtaining Raman spectra where both the composition and physical–chemical approximations are fully known, enabling the mapping of structures and their corresponding spectra. Combining theory and experiment is therefore a valuable strategy to achieve a comprehensive picture of the behavior of nanomaterials. Theoretical Raman spectra in combination with experimental spectra have been used successfully in the past by several groups, including ours [15,16,17,18,19].

Raman has been extensively used to characterize titania-based materials. For instance, Wachs and coworkers used Raman to characterize catalytic mixed oxides where titania can be the active phase or the support [20]. Typically, the titania catalysts were identified by their fingerprint region between 300 and 700 cm^−1^ [21]. In the case of TiO_2_ nanoparticles, their properties strongly depend on their size, shape, and exposed facets [22] which, in turn, are affected by the synthesis conditions. Moreover, it has been found experimentally that crystalline TiO_2_ nanoparticles prefer the anatase structure for diameters up to about 10 nm, and above this size they crystallize in a rutile structure. Raman spectroscopy has been highly valuable in evaluating the relative content of distinct TiO_2_ phases in the synthesized nanoparticles, based on the simple interpretation of a spectrum as a superposition of the single-phase spectra [23,24].

Moreover, Raman spectroscopy was used to measure the percentage of exposed (001) facets in anatase TiO_2_ [25]. Wu et al. [26] applied Raman, among other techniques, as a tool to confirm the presence of oxygen vacancies on (101) and (001) facets of anatase samples calcinated in different atmospheres. Micro-Raman mapping was used to characterize an anatase TiO_2_ single crystal with a large percentage of reactive (001) facets [27]. It was observed that the Raman spectra of the crystal varied with the measurement position, i.e., when the (001) or (101) facet was probed, and the measured differences resulted from the orientations and the symmetry rules. Furthermore, the differences between points of the same crystal facet under the same measurement condition were attributed to defects on the surfaces, such as oxygen vacancies, local lattice disorder, etc.

Interestingly, most of the applications rely on empirical assignments of the Raman spectra. From the theoretical perspective, the Raman spectra of nanocrystals were interpreted usually in terms of the quantum confinement effects [23,28,29]. However, phonon-confinement models do not capture all the spectral features. To the knowledge of the authors, theoretical calculations of Raman spectra for different facets of anatase-based nanocrystals are lacking and would clarify the interpretation of experimental data, specifically as regards the origin of the signal, and both the position and the intensity of the peaks. In the present work, we perform a theoretical *ab initio* investigation of the Raman response for a set of TiO_2_ models, selected to identify the main features responsible for the Raman spectra. We carefully analyze the validity of the computational setting and its effect in the theoretical spectra, by comparison with available experimental observations. We provide a systematic analysis of a series of TiO_2_ systems based on state-of-the-art periodic density functional theory DFT calculations, including laser and temperature corrections. We show how a computational protocol can be used to determine the sensitivity of Raman spectra to surface termination, and the dependence of the computed spectra on the size of the nanomaterial. We show that *ab initio* calculations are powerful tools to assign Raman signals origin, with potential use to predict how different structural features may affect the observed spectra.

## 2. Methods and Models

The calculations were performed with the CRYSTAL17 [30] code using localized all-electron basis sets, in 0D, 1D, 2D, and 3D periodic boundary conditions (PBC). The benchmark of different basis sets and exchange–correlation functionals was calculated for the well-studied bulk anatase phase; the details are discussed in the Results section.

The numerical accuracy parameters when calculating the Coulomb and Hartree–Fock exchange series were set to 7 7 7 7 and 14, respectively. The chosen threshold for the total energy convergence of the self-consisted cycle was equal to 10^−10^ Hartree, and the shrinking value for the reciprocal space sampling was set to 6. The exchange–correlation contribution was evaluated by numerical integration over the cell volume with the default integration grid (XLGRID).

In the case of bulk anatase, full relaxation of lattice cell parameters and atomic positions was performed for each combination of basis set and exchange–correlation functional. Surfaces were modelled by infinite slabs calculated as 2D models in CRYSTAL17. The most stable orientations, (101), (001), and (100) were cut out from the optimized bulk parameters, with variable thickness. The lattice parameters of surfaces were kept fixed to the bulk optimum, and only atoms were allowed to relax. The convergence criteria for atomic gradients that are recommended for vibrational calculations (tighter than default ones) were adopted. After optimization, the vibrational frequencies at the Gamma point were evaluated within the harmonic approximation as the eigenvectors of the mass-weighted Hessian matrix [31,32].

Raman intensities were computed by means of a fully analytical approach that is based on a Coupled–Perturbed Hartree–Fock/Kohn–Sham (CPHF/KS) [33,34] approach for the response of the electronic density with respect to the external electric field. Raman scattering intensity due to a *Q_i_* vibrational mode of frequency *ω_i_*, associated with an *xy* component of a polarizability tensor of an oriented single crystal is given by:$$I_{xy}^{i}\propto \left(\frac{\alpha_{xy}}{\partial Q_{i}}\right)$$

The powder (polycrystalline) Raman spectrum can be obtained by averaging over the possible orientations of the single crystal [35].

More realistic conditions can be included by applying temperature and laser correction that is included in the *C* pre-factor in the expression below:$$I_{xy}^{i}\propto C\left(\frac{\alpha_{xy}}{\partial Q_{i}}\right)$$
where *C* depends on the laser frequency *ω_L_* and the temperature *T*: $$C\sim {\left(\omega_{L}-\omega_{i}\right)}^{4}\frac{1+n(\omega_{i})}{30\omega_{i}}$$

The term $n(\omega_{i})$ is the Bose occupancy factor, defined as
$$1+n\left(\omega_{i}\right)={\left[1-exp\left(-\frac{\hbar \omega_{i}}{K_{B}T}\right)\right]}^{-1}$$

The integrated intensities were normalized to the highest peak that is arbitrarily set to 1000. The convolution of the spectra was carried out automatically by CRYSTAL17 with default parameters. Vibrational modes were visualized with CRYSPLOT (https://crysplot.crystalsolutions.eu/index.html).

## 3. Results

### 3.1. Computational Benchmark on Bulk Anatase and Effect of Laser Frequency and Temperature

The bulk anatase structure was used to evaluate the effect of the functional and basis set on geometrical and vibrational properties. Anatase has a tetragonal, body-centered Bravais lattice and belongs to the *I4/amd* space group. The conventional unit cell of anatase is composed of 12 atoms (two primitive cells). Ti atoms are sixfold-coordinated by O atoms, and the resulting structures are chains of distorted TiO_6_ octahedra with two long (axial) and four short (equatorial) Ti–O bonds. Each of the O atoms is then surrounded by three Ti atoms with two equatorial and one apical bond. Figure 1 displays the bulk and selected termination geometry.

Extensive studies on structural and electronic properties of anatase and rutile forms of TiO_2_ calculated with different exchange–correlation functionals were conducted by F. Labat et al. [36]. The authors compared results obtained from two different approaches, one based on plane waves and the other on all-electron Gaussian-like functions. It was pointed out that, in the case of a Gaussian-like basis set, the *d*-polarization function has to be used for both Ti and O atoms to better describe the structural parameters of anatase. Therefore, here, we focused only on two basis sets, DZVP and TZVP, both of which are available in the CRYSTAL17 database, and each of which contains *d*-polarization functions for Ti and O, and, additionally, *f*-polarization functions for Ti atoms only. The two basis sets were tested with selection of exchange–correlation functionals. The optimized lattice parameters and bond distances are summarized in Table 1. The agreement with the experimental data is good, and computed structural parameters are within less than a few percent of the experimental values, whichever basis set or functional was considered. In fact, the PBEsol functional in combination with the TZVP basis for both Ti and O gives a perfect match between the lattice parameters and the experimental ones. PBEsol is designed to improve upon PBE in terms of equilibrium properties of bulk solids, and their surfaces and the lattice constants are systematically lower and therefore better than PBE by 1–2% [37]. Moreover, the bond distances obtained with PBEsol match the experimental values (measured at 15 K [38]), while PBE0 underestimates these values and all other functionals overestimate them.

Group theory predicts six Raman-active vibrations for anatase, 3E_g_ + 2B_1g_ + A_1g_, that are measured at 143 cm^−1^ (E_g_), 198 cm^−1^ (E_g_), 395 cm^−1^ (B_1g_), 512 cm^−1^ (B_1g_), 518 cm^−1^(A_1g_), and 639 cm^−1^ (E_g_). The E_g_ modes are symmetric stretching of Ti–O bonds, while the B_1g_ and A_1g_ are symmetric and antisymmetric bending motions of O–Ti–O, respectively. A schematic representation of anatase vibrational modes and the atoms involved is present in the Appendix A.

Table 2 shows the vibrational frequencies obtained with different computational settings. Our results show that the frequency depends more on the exchange–correlation functional than on the basis set. Note, however, that we only discuss two similar basis sets, and this statement should not be taken as a general rule. The tests we conducted with smaller basis sets for Ti and O were less satisfactory in describing the vibrational properties and were not included here.

It can be seen that the PBE and PBE-D3 functionals systematically underestimated calculated frequencies, and the theoretical values can be shifted by up to ~33 cm^−1^ from the experimental ones. The best agreement with the experiment was obtained with the hybrid B3LYP functional, while the error of the other hybrid functional, PBE0, was bigger, in particular for the two B_1g_ and A_1g_ modes. The PBEsol functional performed better than PBE or PBE-D3, but not as well as B3LYP.

Figure 2a displays the theoretical Raman spectra of anatase obtained with different Hamiltonians. Maschio et al. [34] showed that the calculated intensities are sensitive to the choice of the basis set (and less so to the exchange–correlation functional); however, in our case, the relative peak intensities were rather robust with respect to the computational setting. Nonetheless, with a standard calculation, the theoretical intensities did not match the experimental data [40]. We used the temperature and laser correction for intensities, as implemented in CRYSTAL17, and the relative intensities agree much better with the experimental data (Figure 2b). 

In summary, the benchmark for the bulk anatase showed that structural parameters and bond distances are very well described by the PBEsol functional in combination with TZVP for Ti and O atoms, while vibrational frequencies are closer to the experimental results when the B3LYP functional is adopted with either the DZVP or TZVP basis set. The overall shape of Raman spectra and relative peak intensity are rather robust with respect to the computational settings; however, a temperature and laser correction has to be included to properly match experimental intensities.

In the following, for the calculation of 2D anatase surfaces, we decided to employ the PBEsol functional with the TZVP basis set for Ti and O. Given the number of structures studied, calculations with B3LYP would be cumbersome and would not necessarily lead to a better understanding of physical properties and interpretation of the Raman spectra. The general conclusions should not be affected by the computational setting.

### 3.2. Analysis of Computed Raman Spectra of Selected Surfaces

It is expected that the properties of nanosized materials depend critically on the size, and that at large enough sizes, properties converge to the bulk material. We modelled the effect of size by increasing the slab thickness of the most stable stoichiometric terminations of anatase, namely (101), (001), and (100) [41,42,43], and we studied the evolution of the associated Raman spectra. The lattice parameters and thickness of the slabs considered are listed in Appendix A. The number of monolayers (ML) was counted based on the number of planes containing Ti atoms with different *c* coordinate values. This is rather obvious for (100) and (001) terminations, where titanium planes are clearly separated. Attention should be paid to the (101) termination, where the Ti atoms, e.g., in the surface layers, are very close together along the *c* direction, but since their *c* coordinates are not the same, they are counted as belonging to two different planes. During the structure optimization, the *a* and *b* lattice parameters were kept fixed (equal to the optimized bulk anatase), and only atoms were allowed to move. We calculated the surface energy for each of the slabs, they are shown in Appendix A. In accordance with previous theoretical findings, the most stable termination is (101), while the least stable is (001). For the (101) and (001) surfaces, the surface energy converges to some value. For the (100) termination, the surface energy oscillates depending on the parity of the number of layers in the slab (Appendix A).

Anatase surfaces are characterized by the presence of undercoordinated titanium or oxygen atoms, and it is expected that the Raman spectra are sensitive to the surface details. Figure 1 displays the characteristic coordination of the terminations selected. In general, the Raman modes that are active in the bulk anatase should be also active in thin films/surfaces (although the opposite is not always true). Our results show that the “bulk” modes can be found in surface structures, but the decrease in the symmetry of the system from 3D to 2D implies different coordination of the surface atoms with respect to the bulk, leading to changes in the spectra. For simplicity, we will keep in the following the same nomenclature as for the bulk to describe the bulk-like modes in the surface slabs, and we will analyze in detail the origin of the new peaks for (101), (001), and (100) terminations.

#### 3.2.1. The (101) Termination

The (101) termination has a characteristic sawtooth profile and consists of fivefold-(Ti_5c_) and sixfold (Ti_6c_)-coordinated titanium atoms, as well as twofold (O_2c_) and threefold (O_3c_) oxygen atoms as schematically presented in Figure 1a; specific labelling and coloring for Raman assignment is shown in Figure 3. For optimized surfaces we found that the fully coordinated Ti2 atoms at the surface move outwards, causing theTi2-O5 bonds to elongate to ~2.080 Å (the initial equatorial bond was 1.931 Å), while the undercoordinated Ti1 atoms displace inwards, causing the Ti1-O4 bonds to shorten to ~1.780 Å (from an initial 1.930 Å). The surface Ti–O_2c_ (blue O) bonds also experience important shortening, and we observed Ti1-O1 bonds of ~1.820 Å and Ti2-O1 bonds in the range of 1.840–1.850 Å. All bond distances for the considered (101) slabs can be found in the Appendix A. Here, it is important to note that the Ti–O bond distances in the surface layers do not change much (less than 0.01 Å of difference) between slabs with different thicknesses. The Ti_6c_ (grey)-O_3c_ (pink) distances are only slightly modified in the layers close to the surface (the 2nd and 3rd layers), and then converge fast to the bulk values. From the 4th to 5th middle layers, the Ti_6c_-O_3c_ (pink) bond distances resemble the bulk anatase, indicating that at least 10 ML would be required for a slab mimicking a realistic surface. On the other hand, the Ti_6c_-O_3c_ (red) distances are more affected by the presence of the surface, and a slab of at least 16 ML would be needed to find Ti_6c_-O_3c_ (red) distances close to the bulk values in the middle of the slab (bulk-like region).

Figure 3 compares the evolution of the calculated Raman spectra with the thickness of the (101) slabs. For convenience, frequencies of bulk anatase modes are indicated by vertical dashed lines.

As expected, with increasing slab thickness, the Raman spectra of slabs converges to those of the bulk. For instance, the most intense peak, found at 151.9 cm^−1^ for the 6 ML slab, shifted to 144.3 cm^−1^ for the 20 ML slab, and corresponds to E_g_(1) bulk mode with a frequency of 143.8 cm^−1^. A similar example is the Raman mode at 622.6 cm^−1^ for 6 ML, which moved to 624.9 cm^−1^ for the 20 ML slab and experienced a gradual increase in intensity with increasing slab thickness.

To better understand the evolution of Raman spectra and the origin of the peaks in the slab models, we used CRYSPLOT to track the atoms that are involved in a particular vibration. Since the number of calculated frequencies depends on the number of atoms in the structural model and increases with slab thickness, we limited our analysis to the most intense peaks present in the spectra. Visualization of the vibrational modes allowed us to define frequency regions dominated by the motion of oxygen or titanium atoms and to distinguish signals from atoms with different coordination or positions within the slab; these regions are marked in Figure 3. As can be seen, the higher-frequency region (above 400 cm^−1^) is dominated by pure oxygen vibrations, while in the lower-frequency part, pure titanium modes can be found along with mixed modes that couple motion of oxygen and titanium.

The oxygen atoms with different coordination present distinct Raman bands. It is therefore possible to separate signals from O_2c_ (blue) or O_3c_ (orange) surface atoms from the O_3c_ (red and pink) atoms in the bulk-like region of the slab. For example, the two peaks around 600 cm^−1^ are both pure oxygen vibrations that correspond to the Ti–O stretching mode (E_g_(3)), but the higher peak at ~625 cm^−1^ is related to O_3c_ (pink) atoms from the middle layers, while the peak with a lower frequency at ~601 cm^−1^ is due to the O_3c_ (orange) atoms in the surface layer. Moreover, the small peak at ~730 cm^−1^ is linked to the vibrations of surface O_2c_. The differences in the calculated frequencies are associated with variations in the Ti–O bond lengths and the corresponding force constants that in turn affect the frequency value. For the optimized slabs, the Ti–O_3c_(orange) bond lengths were longer than in the inner layers, which results in the lower vibrational frequency of O_3c_ (orange) atoms. The Ti–O_2c_ bonds are ~1.8 Å, and are much shorter than O_3c_ (surface or bulk) (~1.95 Å), which causes a higher vibrational frequency. The relative intensities of the O_3c_ (bulk, pink) and O_3c_ (surface, orange) peaks change with the slab thickness and the O_3c_ (bulk) signal becomes dominant as the slab increases in size and suppresses the signal of the surface atoms. The shifts in the O_3c_ (orange) and (pink) peaks when slab thickness increases from 6 ML to 20 ML are minor (~2–3 cm^−1^ of difference), and are related to small variations in the associated Ti–O distances. An example of a visual representation of the chosen modes for a (101) with a 12 ML slab, along with a table showing the frequencies and their intensities, can be found in the Appendix A, respectively.

The Raman bands that are visible at 500 cm^−1^ (B_1g_(2)) and 515 cm^−1^ (A_1g_) for the bulk anatase are also present in the spectra of slab structures. As before, the thicker the slab, the better the agreement between the surface and bulk signals. In the case of the slabs, we found a few modes with similar frequencies and varied intensities in this region that are associated with vibrations of O_3c_ (pink and red) atoms from the inner and surface layers (O_3c_ orange). The modes of the 12 ML slab are depicted in Appendix A. The motion of oxygen atoms in the slabs is similar to the bulk, and corresponds to the O–Ti–O bending mode.

The B_1g_(1) mode, that was expected to be ~381 cm^−1^ for the anatase, was found at much higher frequency (~430 cm^−1^) in the 6 ML slab and converged gradually to the bulk value when the slab thickness increased. Visualization of the vibrating atoms revealed that, in contrast to the bulk B_1g_(1) mode, in which only titanium is moving, the motion of titanium and oxygen atoms is coupled in the case of (101) slabs. Moreover, the surface Ti and O atoms (mostly O_3c_ (orange) with a minor part of O_2c_) are also involved, and their contribution decreases slowly with slab thickness.

The last mode of anatase E_1g_(1) is clearly present in all of the (101) slabs, and only the Ti atoms move in the bulk. Again, we found a few modes with frequencies ~144 cm^−1^ that involve Ti atoms from different inner layers of the slabs, as illustrated in Appendix A. The higher frequency peak (shoulder) of the E_1g_(1) mode is associated with coupled Ti and O vibrations of atoms close to and in the surface layers. The intensity of the shoulder peak decreases with the slab thickness, while the bulk-like peak increases at the same time.

The remaining modes that are present for (101) slabs but not for the bulk anatase appear to be due to the reduction in symmetry from 3D to 2D. They have much lower intensities and will not be discussed.

Note that for all of the (101) slabs there was one negative frequency appearing in the calculations that would suggest possible instability in the structures. By default, the calculations with CRYSTAL17 exploit the space group symmetries, and during structure optimizations, the symmetries are kept fixed, which in turn restricts possible atomic displacements. As we show, after removing the symmetry constraints and re-optimizing the slabs, the negative frequency was no longer present. The difference in the optimized structures and the associated Raman spectra with or without symmetry constraints is minor (see Appendix A).

#### 3.2.2. The (001) Termination

The most reactive bulk-cut (001) termination was rather flat, and only fivefold-coordinated Ti atoms (Ti_5c_) could be identified, whereas twofold- (O_2c_) and threefold (O_3c_)-coordinated O atoms are present at the surface.

After structural optimization, the surface Ti_5c_-O_2c_ (Ti1-O1) bonds were elongated by less than 1% with respect to the initial equatorial bonds, and both of them had the same length, equal to 1.950 Å. The Ti_5c_-O_3C_ (Ti1-O2—equatorial) bonds, on the other hand, were shortened to 1.922 Å. The apical bonds between Ti1-O3 and Ti2-O2 were also contracted to ~1.910 Å and ~1.960 Å, respectively. All other atom distances were not affected by the presence of the surface, and were equivalent to the bond lengths in the bulk anatase (see Appendix A).

Figure 4 displays the evolution of the Raman spectra for the (001) facet. The spectra clearly differ from the (101) termination and, at first glance, do not seem to converge to the bulk anatase, even for the 24 ML slab. A closer examination shows that modes of the bulk anatase can be identified in the (001) slabs as well (marked by dashed lines), but their relative intensities are inversed with respect to the bulk spectra. The E_1g_(1) and E_1g_(3) modes, which are the most intense bands for the bulk anatase, have the smallest intensities for the (001) slabs, while the weak bulk modes B_1g_(1), B_1g_(2), and A_1g_ become the most intense for the thickest (001) slabs. We checked that the E_1g_(1)-like mode of ~144 cm^−1^ is related to the movement of Ti atoms from the middle layers of each slab in a plane parallel to the surface, which corresponds to a similar type of activity in the bulk anatase and is associated with Ti–O stretching vibration. The E_1g_(3) mode at ~625 cm^−1^ is also identified as the Ti–O stretching mode, but this time it is related to the movement of inner O_3c_ atoms in a plane parallel to the surface. An example of visualization of the chosen modes for the 12 ML (001) slab is shown in the Appendix A. Both E_1g_(1) and E_1g_(3) peaks do not change much with slab thickness, and the frequency variation between different thicknesses is less than 2–3 cm^−1^ and is followed by negligible variations in intensity.

The B_1g_(2) (~500 cm^−1^) and A_1g_ (~515 cm^−1^) modes are associated with movement of O_3c_ atoms along the *c* direction and correspond to the O–Ti–O bending mode in slabs and in the bulk anatase. As before, in the case of the slabs, it mostly involves O_3c_ atoms from the inner layers, and the contribution of O_3c_ close to the surface is smaller and decreases when the slab gets thicker. The frequency of these modes changes between the thinnest and thickest slabs by about 10 cm^−1^, and their intensity increases significantly (the intensity of B_1g_(2) mode for 24 ML is almost four times higher than in the 6 ML slab, and that of the A_1g_ mode is seven times more). The strong band around 530 cm^−1^ that is visible for (001) structures is related to the motion of O_3c_ atoms (along the *c* direction) from the surface and close-to-surface layers, which implicates Ti1-O3 and Ti2-O2 bonds. The higher frequency of the surface-related mode is due to the Ti1-O3 and Ti2-O2 bonds being shorter with respect to the inner ones. The intensity of the 530 cm^−1^ peak, however, decreases very slowly with the slab thickness, and for the considered slab thicknesses it is not negligible.

The band corresponding to the B_1g_(1) mode shifts significantly between 6 ML and 24 ML slabs (almost by 40 cm^−1^), and is accompanied by an increase in intensity. In the bulk anatase, the B_1g_(1) mode is due to the movement of Ti atoms along the *c* axis (see Appendix A), and a similar type of motion was expected for the slabs. Indeed, we found that the most significant contribution comes from Ti atoms (except for Ti in the surface layer) moving along the *c* direction; however, oxygen atoms are also involved. Moreover, the involvement of the oxygen network is more important when the slab thickness decreases.

A new peak that emerges around 421 cm^−1^ originates from the coupled movement of O and Ti from the surface layer (O_2c/3c_ and Ti_5c_) and its proximity (O_3c_ and Ti_6c_). The atoms vibrate along the c direction, and the biggest contribution comes from surface O_2c_ atoms. The intensity of this peak decreases slowly with slab thickness.

The modes found below 100 cm^−1^ include collective motion of the whole lattice (although some Ti/O vibrations can be identified), and these modes will not be discussed. The calculated Raman frequencies and intensities are shown in Appendix A.

It is important to note that, with a “standard” calculation in CRYSTAL17, for each (001) slab, two negative frequencies were present with values over −300 cm^−1^ that correspond to Ti–O stretching vibration of surface Ti_5c_ and O_3c/2c_ atoms. Re-optimization of the slabs without symmetry constraints led to a different surface geometry, i.e., the two Ti_5c_-O_2c_ surface bonds were no longer equivalent; one shorter and one longer Ti–O bond was found (see Appendix A). Such surface relaxation has already been reported in previous theoretical works, and was found to be the most stable configuration for (001) termination [43,44]. Figure 5 illustrates the evolution of Raman spectra of the “non-symmetric” (001) surface, which clearly differs from the Raman spectra of the “symmetric” (001) surface. Nonetheless, certain modes resembling those found in the bulk anatase can still be identified, such as E_g_(1), B_1g_, and A_1g_, and were ascribed to a similar type of motion as in the bulk anatase. As for the ”symmetric” (001) structure, the B_1g_(1) mode shifts the most with the slab thicknesses ranging from 6 ML to 14 ML.

The peaks associated with the E_g_(3) mode exhibit a more complex behavior. Depending on the number of layers in the slab, either one (8 ML/12 ML) or two (6 ML/10 ML/14 ML) intense peaks can be identified in the frequency range between 600 and 700 cm^−1^. Among these, the peak at approximately 654 cm^−1^, present for -6, 10, and 14 ML (marked by arrow number (4)), remains relatively stable across slabs with different thicknesses. Our analysis using CRYSPLOT revealed that this peak is associated with the collective motion of oxygen network, mainly involving surface O_2c_ atoms. In fact, a similar type of motion with similar frequency was also observed in the 8 and 12 ML slabs, with significantly lower intensity. It appears that the differences in intensity for this mode might be attributed to the particular geometry of slabs. As schematically illustrated in Figure 5, slabs with 6, 10, and 14 ML are symmetric with respect to a mirror plane, while for 8 and 12 ML slabs, this symmetry is absent, which in turn potentially affects dipole moments and the polarization in the structures. The remaining peaks around 625 cm^−1^ are attributed to O atoms from the inner layers of the slabs, and the intensity of these peaks increases with slab thickness. The intensity of the E_g_(3) mode dominates the spectra, which differs from the case of the symmetric slab or the experimental spectra. The intensity of this mode appears to gradually decrease for the 14 ML slab, and may further decrease with increased slab thickness. However, attempts to perform structural optimizations for thicker slabs were unsuccessful, and it is not possible to make clear predictions at this point.

The remaining new modes, highlighted by arrows (1), (2), and (3), are dominated by surface and near-surface oxygen atoms. Surprisingly, the intensity of these modes increases with increasing slab thickness and remains clearly present even in the thickest slab considered. To ascertain whether this intensity increase is not due to applied temperature/laser correction, we plotted the evolution of Raman spectra without the correction, as shown in Appendix A. The result confirms that the intensity of peaks (2) and (3) indeed increases with the slab thickness, while the intensity of the peak (1) is relatively small and exhibits small variation among the slabs. Currently, the exact cause of this different behavior is unclear, and it is possible that the surface peaks may eventually be dominated by growing bulk-like modes. This example highlights the sensitivity of Raman spectroscopy to structural changes and the underlying structural model, emphasizing the importance of addressing the surface reconstruction of anatase (001) in future studies.

#### 3.2.3. The (100) Termination

The (100) surface layers form steps where the outermost Ti atoms are fivefold-coordinated (Ti_5c_) and the Ti atoms in the second layer, that are also exposed, are sixfold-coordinated (Ti_6c_). Both twofold- (O_2c_) and threefold (O_3c_)-coordinated oxygen atoms are present on the topmost layer, and only threefold-coordinated (O_3c_) oxygen atoms can be found in the second layer.

The structural optimization shows that surface O_3c_ (orange) atoms shift upwards, while Ti_5c_ atoms shift downward. This leads to the elongation of surface Ti_5c_-O_3c_ bonds with respect to bulk anatase. The final Ti2-O2 (Ti1-O1) and Ti2-O1 (Ti1-O2) bonds increase to about 2.06 Å and 1.96 Å, respectively. The bridging O_2c_ atoms shift towards Ti surface atoms to shorten the Ti_5c_-O_2c_ and Ti_6c_-O_2c_ bonds to ~1.820 Å and ~1.850 Å, respectively. The Ti_5c_-O_3c_(red) (or Ti2-O7/Ti1-O8 in Figure 6) are also strongly affected, and are shortened to ~1.80 Å. The Ti_6c_(surface)-O_3c_(red) bonds (Ti4-O11/Ti3-O12), on the other hand, are elongated to ~2.04 Å.

The Ti_6c_-O_3c_ (pink and green) distances are close to the bulk anatase values of ~1.930 Å and 1.980 Å, and do not change significantly between slabs with different thicknesses. As for the (101) termination, the Ti–O_3c_(red) bond lengths are more affected by the presence of the surface. All bond distances can be found in Appendix A.

Interestingly, different trends were observed for the optimized Ti–O bond distances depending on the parity of the number of layers and the slab thickness. For instance, the Ti1-O2 (or Ti2-O1) distance in the 6 ML slab starts from 2.04 Å and increases to 2.06 Å in the 14 ML slab, while in the 5 ML slab it is 2.08 Å and decreases to 2.06 Å in the 13 ML slab. Other examples of such variations in Ti–O distances can be found in Appendix A. It is not the scope of this paper to investigate in detail such structural behavior, but it is important to say that these differences will have a direct impact on the Raman spectra. Note that slabs with an “odd” number of layers are symmetric with respect to a mirror plane that bisects the slabs and passes through the middle layer, which is not the case of the slabs with “even” number of layers.

The evolution of the Raman spectra for (100) slabs with “odd” and “even” numbers of layers are presented in Figure 6. As expected, the surface spectra converge to the bulk spectra for slabs at a certain thickness.

The E_g_(1) mode in (100) slabs can be attributed to the vibrations in middle Ti atoms in a plane parallel to the surface. It resembles the type of motion present in bulk anatase; however, the mode frequency and its evolution depends on the layer’s parity. In slabs with an odd number of layers, the E_g_(1) mode starts with a lower frequency of ~121 cm^−1^ for the 5 ML structure and shifts to ~142 cm^−1^ for the 13 ML slab, whereas in slabs with an even number of layers, the E_g_(1) is found initially at a higher frequency of ~148 cm^−1^ in the 6 ML slab and moves to ~143 cm^−1^ in the 14 ML slab. Such behavior of the E_g_(1) mode follows changes in associated Ti_6c_-O_3c_(pink/green) bond distances and reflects different trends in bond lengths for the “odd” and “even” slabs discussed earlier.

The E_g_(3) mode at ~625 cm^−1^ is ascribed to the Ti–O stretching vibration, which, in (100) slabs, originates from the movement of O_3c_ (pink and green) atoms in a plane parallel to the surface; however, the contribution of each oxygen type depends on the slab “parity” and the position of atoms within the slab. Visualization of vibrational modes for 11 ML and 12 ML slabs is presented in the Appendix A, and illustrates the differences in O_3c_ (pink and green) contribution for slabs with “odd” and “even” numbers of layers. A detailed analysis of evolution of each vibration in this region would be cumbersome and probably confusing, thus it should be enough to note that, when the slab thickness increases, the signal from the O_3c_ (pink and green) atoms from the inner layers becomes dominating and approaches bulk E_g_(3) frequency. The signal of the O_3c_ atoms (orange) from the surface layer is found at ~577 cm^−1^ for the 5 ML slab (~596 cm^−1^ for 6 ML), and shifts to ~591 cm^−1^ for the 13 ML slab (~592 cm^−1^ for 14 ML) with decreasing intensity.

The B_1g_(1) mode ~388 cm^−1^, associated with Ti motion in the bulk anatase, can be identified in slabs as well; however, in the slab, movement of Ti atoms is coupled with the movement of O atoms. The signal comes mostly from Ti atoms in the middle layers, but the contribution of O_3c_ atoms from the surface can also be observed (see Appendix A with mode visualization for 12 ML slab). As the slab thickness increases, the participation of inner Ti atoms becomes greater than that of oxygen network and the mode converges slowly towards the bulk spectra; for the 12–14 ML slabs, the B_1g_(1) can be found at ~388 cm^−1^.

The B_1g_(2) and A_g_ modes can be easily identified in the slab spectra and seem to converge to bulk-like values much faster than the B_1g_(1) mode; already for the 6 ML slab the peaks are found close to those of the bulk modes, with correct relative intensities. As in a bulk, these modes are related to the motion of oxygen atoms in the inner layers of the slabs and can be interpreted as O–Ti–O bending vibration.

As before, peaks appearing in the (100) Raman spectra and not present for the bulk anatase can be ascribed to the presence of surface and the reduction in symmetry from 3D to 2D. For example, the O_2c_ atoms contribute to the modes found ~700 cm^−1^; however, this time it is not an isolated vibration, but some of other O_3c_ oxygen atoms that are contributing to the vibration. The intensities of these modes is negligible with respect to other peaks in the spectra. The calculated Raman frequencies and intensities are in Appendix A.

## 4. Discussion

### 4.1. Raman Spectra Are Sensitive to Slab Thickness

Theoretical study of the Raman spectra for anatase-based surfaces has shown, as expected, that for each of the considered terminations, the spectra converge to the bulk spectra as the thickness of the slabs increases, and the main bulk-like modes can be identified. In Figure 7, we present the evolution of the difference between frequencies of the main modes in the (101), (001), and (100) terminations and the bulk as a function of slab thickness. Note that, with increasing number of atoms (i.e., number of layers in the slabs), the number of calculated frequencies increases and more modes with different relative intensities are found in regions associated with the main bulk modes (these modes are summarized in Appendix A). As a consequence, it was difficult to distinguish between B_1g_(2) and A_1g_ modes, which appear close to each other and are also hard to separate experimentally. Therefore, we decided that, if peaks were in the range of 2–3 cm^−1^, then the corresponding frequency was obtained as a weighted average of these peaks. Otherwise, if the peaks were further away, the mode with the highest intensity was taken. In the case of the (100) termination, the identification of the B_1g_(1) and B_1g_(2)/A_1g_ modes was not straightforward either, because many modes were present in the middle frequency range (300–600 cm^−1^), especially for thinner slabs (see Figure 6). It was necessary to trace the atoms that are involved and type of motion for each mode, in order to choose the proper frequencies. Due to these difficulties, it is expected that the error made for B_1g_/A_1g_ modes is higher than for E_g_ modes.

We found that, for all of the terminations, the E_g_ modes were rather robust and converged easily to the bulk frequency. A thickness of 2.5–3 nm is required for (101) and (001) terminations. The frequency change for B_1g_(2)/A_1g_ was bigger than for the E_g_ modes; nonetheless, these peaks also converged fast to bulk-like values. The B_1g_(1) mode, on the other hand, showed the most significant dependence on the slab thickness for all of the considered terminations. In contrast to a bulk anatase, where this mode originates from the vibration of the Ti atoms, the motion of Ti in the slabs is coupled with the movement of O, and the participation of the surface oxygen atoms is also present. The oxygen involvement increased for thinner slabs, which might be one of the reasons why the B_1g_ mode depends so strongly on the slab thickness. It is also possible that this shift comes from polarization/dipole effects, and that more layers are needed to properly screen the inner region from the surface.

The difference between frequency values of the main modes in slabs and the bulk decreases with the slab thickness; nonetheless, only in few cases do these values perfectly match (mainly for the E_g_ modes). These inaccuracies might be caused partially by the aforementioned problems with mode assignments in the surface models. Moreover, vibrations in slabs are more complex than in a bulk anatase, and the mode frequency is affected by the position of the involved atoms with respect to the surface and by the coupling of movement of Ti and O networks. All this makes the proper evaluation of the vibrations more complicated and could be a reason of larger numerical uncertainty than in a bulk system.

The relative peak intensity and the overall shape of the spectra converge with frequencies of the bulk spectra. In addition to the main bulk modes, additional peaks appear in slabs due to the lowering of symmetry and the presence of atoms with modified coordination at the surface. However, as can be expected, the contribution of surface vibrations becomes less important with size, and the bulk vibrations govern the overall spectral shape. The spectra of (101) and (100) terminations behave in a similar manner, and for the thickest slabs considered are in good accordance with bulk spectra. We estimate that a good convergence to the bulk spectra for the (101) termination is achieved for slabs with at least 18–20 ML, which corresponds to 3–3.5 nm size. For the (100) termination, 13–14 ML are required, which is equivalent to ~2.5 nm.

The (001) termination is more complex, and even though the peaks associated with surface atoms are still visible in the spectra, a good convergence of the spectra could be obtained for 20–24 ML slabs which correspond to at least 4.5–5 nm thickness.

In summary, size effects can be captured in calculated Raman spectra, as they are sensitive to the amount of surface vs. bulk vibrations for small sizes below 4–5 nm. For larger sizes, the variations are not expected to be significant.

### 4.2. Raman Spectra Are Sensitive to the Surface Termination

In Figure 8, we compare the theoretical Raman spectra for (101), (001), and (100) terminations for the thickest slabs considered with the bulk anatase. The spectra of (101) and (100) terminations are similar and resemble the standard spectra of bulk anatase with dominant E_g_ peaks. Note that in the (101) and (001) terminations, some of the modes associated with surface atoms appear close to E_g_ modes, and even though surface peaks are substantially suppressed for the thick slabs, they do not disappear completely. It is possible that with a further increase in the slabs’ thickness, the surface contribution to the spectra will continue to decrease, but the possibility that those surface modes could lead to broadening of the E_g_ peaks (or tails) in the measured spectra cannot be ruled out at this stage.

In the case of (001) termination, we will consider only the structures with equivalent Ti_5c_-O_3c_ bonds (the “symmetric” slabs). This termination is clearly different to the other two, with the most intense peaks in the mid-frequency range corresponding to the two B_1g_ and A_1g_ modes of the bulk phase. The differences we find between Raman spectra of (101) and (001) terminations agree with experimental data on TiO_2_ nanostructures. Previous experimental studies used Raman spectroscopy to discriminate between different exposed facets in anatase nanocrystals, or to verify the orientation of the TiO_2_ thin films [22,25,27]. In fact, the intensity ratio between E_g_(1) and A_1g_ peaks was used to measure the percentage of exposed (001) facets in the TiO_2_ nanocrystals [25]. In many of the experimental papers, however, the observed differences between (101) and (001) facets were associated with the number of different types of atomic vibrations present for a given facet, including the surface atoms. In other words, since the surface of (101) facets consists of saturated Ti_6c_ and O_3c_ and unsaturated Ti_5c_ and O_2c_, while the (001) facet contains only unsaturated Ti_5c_ and O_2c_ bonding modes on the surface, the percentage of (001) facets would be correlated with the variation in the number of the symmetric stretching vibrations and antisymmetric bending vibrations of O–Ti–O, thus explaining the intensity ratio of the E_g_ and A_1g_ peaks in the Raman spectra [25,45].

Our results clearly show that, due to the reduced coordination of atoms in the surface layer and their reorganization, the vibrational modes of surface atoms have different frequencies and can be distinguished from the bulk-like modes. The difference between (101), (100), and (001) facets come rather from the surface orientation/geometry and determination of the preferred growth direction for facets.

Keep in mind that the (001) termination is a special case, and the surface reorganization plays an important role. It was observed experimentally that the (001) surface can undergo a (1 × 4) surface reconstruction in vacuum conditions, but it was not clear if these surfaces were reconstructed in solution [46,47,48,49,50]. Moreover, to stabilize the (001) facet, the synthesis of TiO_2_ nanostructures is made with using HF that is removed afterwards [25,51,52,53]. Even though previous theoretical studies [43,44] have reported that, for the non-reconstructed (1 × 1) (001) surface, the Ti_5c_-O_3c_ bonds differentiate and become non-equivalent to lower the system energy, there is no clear reason that this should be valid in real conditions where the temperature effects have to be considered. It is possible that, at room temperature, the surface Ti–O bonds are fluctuating, and we should see an average of shorter and longer bonds. Therefore, we decided to consider the case of unreconstructed (001) surface with the two Ti_5c_-O_3c_ bonds equivalent. The Raman spectra of selected “non-symmetric” (001) slabs with non-equivalent Ti_5c_-O_3c_ bonds show much more features than the standard (001) termination and agree much less with the experimental data. They will therefore not be discussed further.

The impact of surface orientation on Raman response is clearly visible when considering independent single crystal directional spectra (Appendix A). Our results, in accordance with reference [39], show that, for bulk anatase, the nonvanishing elements of the Raman tensor for the A_1g_ mode are *xx*, *yy*, and *zz*; for the B_1g_ mode, they are *xx* and *yy*, and for E_g_ modes, they are the *xz* and *yz* components. However, when a surface is formed and symmetry is reduced, the Raman tensor elements are altered in accordance with the surface orientation. Although the single crystal spectra of the (001) termination resemble bulk anatase, the A_1g_ and B_1g_ mode intensities are more pronounced relative to the E_g_ modes. For (101) and (100) terminations, the dominant contribution comes from the *xy* spectra component (to E_g_ modes), which was absent in the bulk phase. Additionally, the total Raman spectra for (101) and (001) terminations have negligible differences, but the single crystal spectra suggest that these orientations can be differentiated. Although differences between *yy*, *zz*, *xz*, and *yz* single crystal spectra are predicted theoretically, the associated peak intensities are low and might be difficult to observe experimentally. Nonetheless, analysis of single-crystal directional Raman spectra confirms that mode intensities depend on structural geometry and orientation.

## 5. Conclusions

In this work, the origin of Raman activity in anatase stoichiometric bulk and surfaces is presented. In particular, a computational protocol is provided to obtain accurate Raman spectra from ab initio calculations. The computed Raman spectra are sensitive to the choice of functional and basis set, and satisfactory spectra (peaks and relative intensities) for bulk and slabs can be obtained using PBEsol/TZVP settings, including laser and temperature corrections with the CRYSTAL17 code.

Additionally, a rooted assignment of the main peaks was carried out based on the thorough analysis of the vibrational modes, connecting the microscopic structure with the observed spectroscopic response. The peaks arising from surface vibrations were identified for the (101), (001), and (100) terminations. The changes in the computed signals with the size, for a range between 1 and 4.5 nm corresponding to a controlled increase in the slab thickness, show that several peaks are extremely sensitive to the thickness, whereas others are robust. This could be used to monitor the size of experimental samples. Moreover, the different terminations can be mapped by selecting specific peaks assigned to each of the facets, whose origin is clearly established in this work. This supports the use of Raman for quantifying the content of a given facet in a sample reported in the experimental literature.

The data generated in this work will serve as a guideline for the experimental characterization of titania-based nanomaterials. They could be applied in a wide range of TiO_2_ materials (thin films, single crystals, facetted nanoparticles, nanosheets…) to assess size and shape effects using Raman. Other conditions (hydration, defects, pressure…) should be addressed in further works to complete the description of other titania systems.

## Figures and Tables

**Figure 1 nanomaterials-13-01856-f001:**
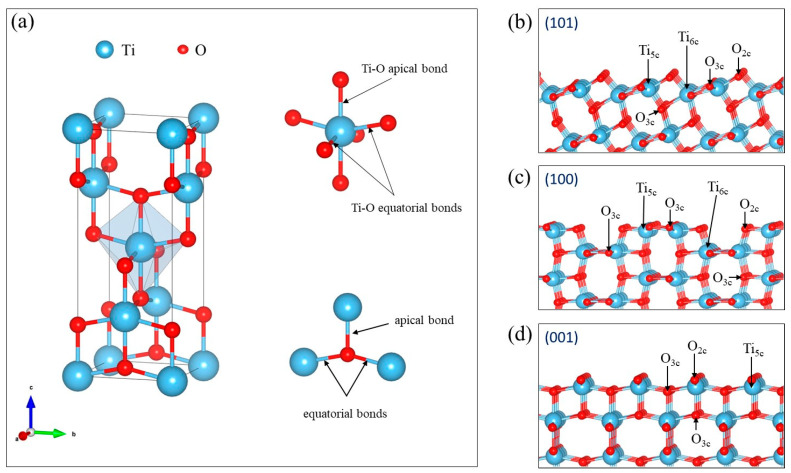
(**a**) Structure of bulk anatase with TiO_6_ octahedron and O_3C_ environment. (**b**–**d**) Three terminations of anatase-based surfaces with the coordination of surface atoms indicated.

**Figure 2 nanomaterials-13-01856-f002:**
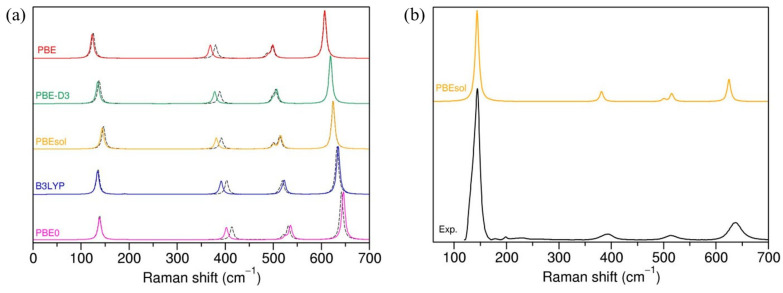
(**a**) Raman spectra for anatase calculated with different exchange–correlation functionals and DZVP (dashed line) or TZVP (solid line) basis sets. (**b**) Raman spectra of anatase calculated with PBEsol and TZVP basis set with temperature and laser correction for intensities (orange) compared with experimental spectra of anatase (taken from RRUFF database, R070582).

**Figure 3 nanomaterials-13-01856-f003:**
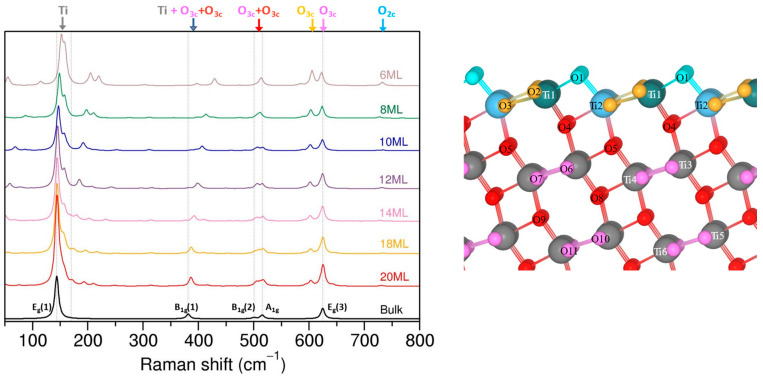
Evolution of Raman spectra for (101) termination with increasing slab thickness. The temperature correction for intensity is included. The spectrum in black is bulk anatase with the corresponding modes nomenclature. The schematic model of the (101) termination shows different types of Ti and O atoms with their specific labels and coloring for Raman peak assignments.

**Figure 4 nanomaterials-13-01856-f004:**
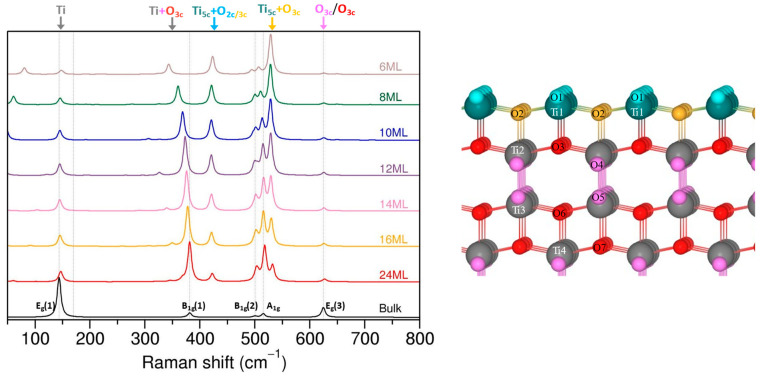
Evolution of Raman spectra for (001) termination with increasing slab thickness. Spectra includes temperature correction for the intensities. The schematic model of the “symmetric” (001) termination shows different types of Ti and O atoms with their specific labels and coloring for Raman peak assignments.

**Figure 5 nanomaterials-13-01856-f005:**
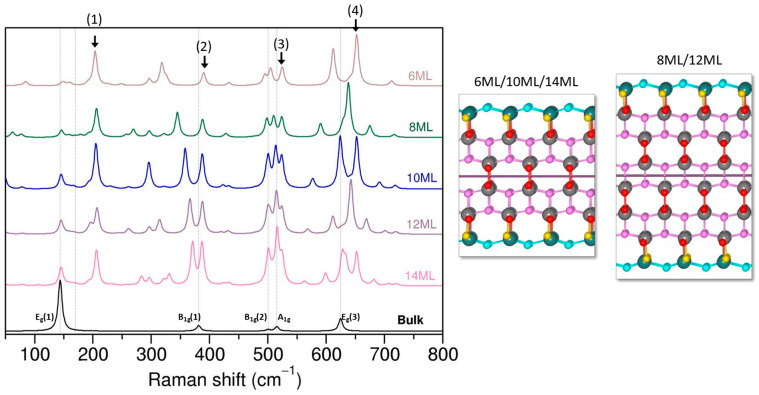
Evolution of Raman spectra for selected “non-symmetric” (001) terminations. The schematic models of the “non-symmetric” (001) terminations depict difference between slabs with and without mirror plane symmetry.

**Figure 6 nanomaterials-13-01856-f006:**
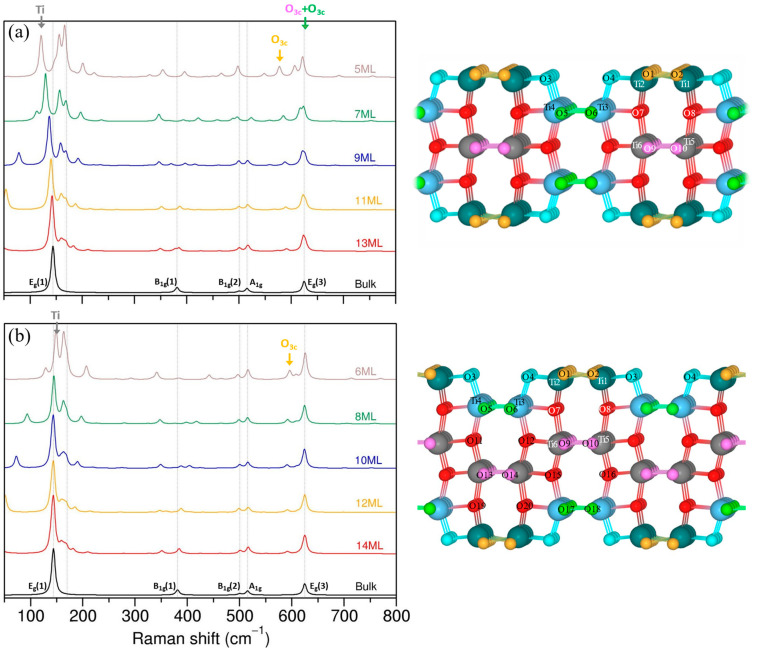
Evolution of calculated Raman spectra for anatase (100) termination for odd- (**a**) and even- (**b**) layer slabs. Temperature correction is included for intensities. The schematic models of odd and even (100) terminations present different types of Ti and O atoms with their specific labels and coloring for Raman peak assignments.

**Figure 7 nanomaterials-13-01856-f007:**
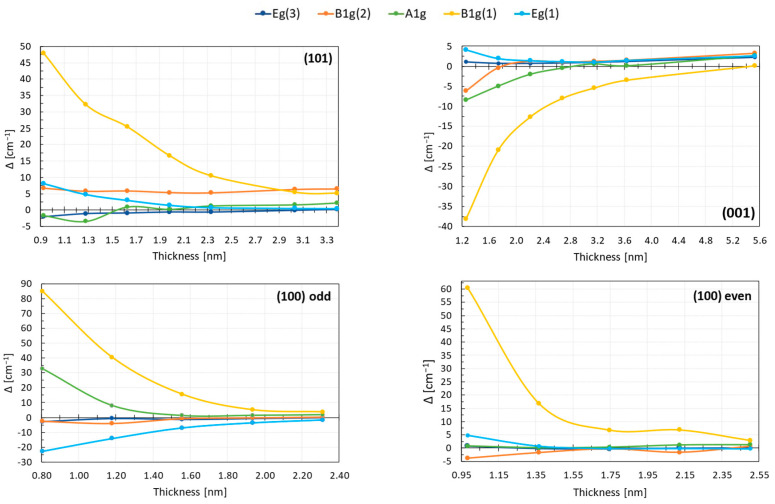
Difference between the frequency calculated for slabs and the bulk value for the main modes as a function of the slab thickness.

**Figure 8 nanomaterials-13-01856-f008:**
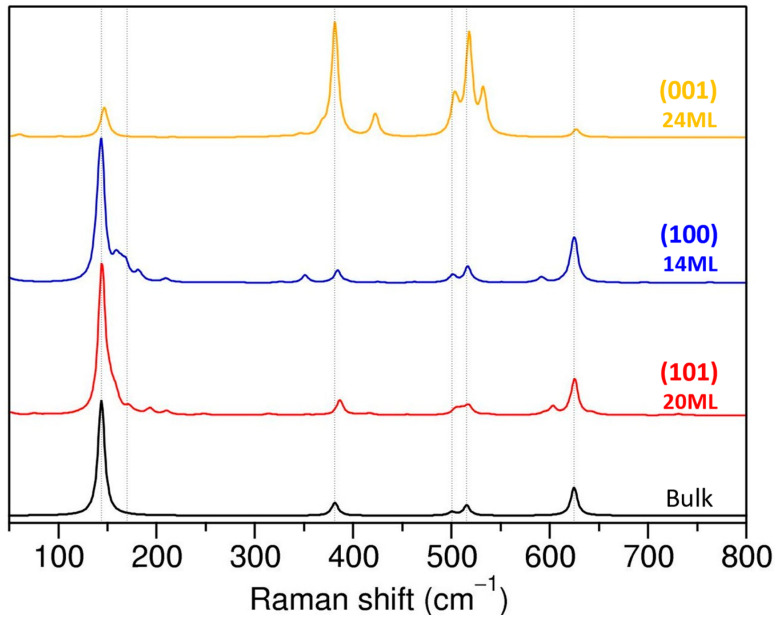
Effect of surface termination in calculated Raman spectra for the slabs.

**Table 1 nanomaterials-13-01856-t001:** Optimized geometry of anatase TiO_2_ calculated with different Hamiltonians. Lattice parameters (*a* and *c*), equatorial (d^eq^) and axial (d^ax^) Ti–O bond lengths of the TiO_6_ octahedra are in Å.

Functional	Basis	a	c	d^ax^	d^eq^
PBE	DZVP	3.809	9.571	1.993	1.946
	TZVP	3.811	9.629	2.001	1.948
PBEsol	DZVP	3.782	9.400	1.973	1.929
	TZVP	3.784	9.463	1.980	1.931
PBE-D3	DZVP	3.799	9.437	1.983	1.936
	TZVP	3.801	9.487	1.990	1.939
B3LYP	DZVP	3.792	9.623	1.987	1.942
	TZVP	3.792	9.665	1.994	1.943
PBE0	DZVP	3.772	9.508	1.971	1.929
	TZVP	3.770	9.536	1.975	1.929
Exp. [38]		3.784	9.515	1.979	1.932

**Table 2 nanomaterials-13-01856-t002:** Raman active modes calculated with different Hamiltonians. All values are in cm^−1^. Vibrational modes are displayed in Appendix A.

		E_g_ (1)	E_g_ (2)	B_1g_ (1)	B_1g_ (2)	A_1g_	E_g_ (3)
Exp	[39]	143	198	395	512	518	639
PBE	DZVP	125.06	179.42	380.01	486.79	497.84	606.67
	TZVP	122.97	174.83	368.84	489.29	499.05	607.22
PBEsol	DZVP	146.51	176.16	391.68	500.15	513.61	624.35
	TZVP	143.79	169.88	381.32	500.40	515.36	624.72
PBE-D3	DZVP	137.40	178.77	388.29	498.17	505.08	618.95
	TZVP	134.63	172.23	377.98	499.01	506.99	619.10
B3LYP	DZVP	135.17	194.66	402.92	512.52	519.09	632.68
	TZVP	134.27	190.30	391.44	517.08	522.58	635.01
PBE0	DZVP	138.45	192.49	413.70	520.11	531.36	641.91
	TZVP	138.48	189.78	402.39	524.54	535.76	646.00

## Data Availability

The input and output files of the calculations for selected structures of each termination are available by NOMAD repository. The data can be found either by dataset name (*CHARISMA-TiO2-Nanomaterials_2023*) or the dataset id number (*Ss-RSA_gROWui9wfjLhNbw*). Additional data are available from the corresponding author, M.C., upon request.

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
