# Peer review of "On the Origin of Raman Activity in Anatase TiO2 (Nano)Materials: An Ab Initio Investigation of Surface and Size Effects"

_nanomaterials, 2023, doi:10.3390/nano13121856_

Round 1

Reviewer 1 Report

The results of this work are highly valuable for a wide range of TiO2 materials (thin films, single crystals, facetted nanoparticles, nanosheets…). Thorough analysis of the vibrational modes function of thickness and terminations was carried out revealing Raman sensitivity to surface details (e.g. termination) and nanomaterial size, never carried before.

My reviewing revealed only strong points related to:

-          Construction of slabs with different thicknesses and comparison with bulk

-          Optimization of geometries by using many functionals and basis sets

-          Labelling of atoms

-          Analysing different surface terminations/projections; selection of surfaces for calculating Raman spectra.

My recommendation is to accept the paper, to be published in Nanomaterials.

Author Response

Responses to the reviewers and list of changes:

  • According to comments of Reviewer 3 we have moved the figure corresponding to spectra of ‘unconstrained” (001) termination from the SI to the main text and we added the small paragraph with proper explanations. For the (101) termination the comparison of Raman spectra for symmetric and not-symmetric structure remained in the SI, since for this termination the effect was rather negligible.
  • Additional information regarding (001) termination was as well added in the SI (Fig. S6 and Tab.S6) and is referenced in the main text of the manuscript (i.e. tale with bond distances and plot of Raman spectra without correction of intensities).

  • The numbering of Figures and Tables was updated in the main text and in the SI accordingly.

The manuscript also shows minor corrections not recommended by Reviewers or Editor:

  • Figures 3-6 were improved (the frequency scale starts now from 50 cm-1). Figure 8 was replaced because in the previous version not correct spectra for (001) termination was chosen (corresponding to 20ML instead of 24ML) (this however does not affect the conclusions or analysis).
  • Information required by Journal was added, i.e. keywords, funding, acknowledgements, conflicts of interest.
  • The format of Bibliography was adjusted to the Nanomaterials format
  • Some typos were corrected.

Reviewer 2 Report

In the manuscript under review, ab initio methods of the density functional theory of the CRYSTAL17 software package, consider the structural features that are responsible for the Raman spectra in the stoichiometric volume of anatase TiO2 and on its (101), (001), (100) surfaces. The origin of the Raman scattering peaks and the effect of structural distortions, laser and temperature effects, surface orientation, and layer size on their positions are analyzed. This can be used to control the size of experimental samples and determine their composition.

It is known that the calculated Raman spectra are sensitive to the choice of functional (PBE, PBE-D3, PBE0, PBEsol, B3LYP) and basis set (DZVP, TZVP). The authors have done a great and fruitful work on the development of the computational protocol. We should agree with the statement that the B3LYP functional provides more accurate values of vibrational mode frequencies in comparison with the experiment, but in relation to large amounts of calculations, the time spent is very large. In this sense, the functionality of PBEsol is more practical. The advantage of the work is that laser and temperature corrections to the intensity of the Raman spectrum are used.

The manuscript and accompanying materials contain a large amount of factual information that may be of use and useful to a wide range of readers. There are no fundamental remarks on the content and design of the manuscript, and it can be published as is.

Author Response

(The authors gave the same response as above.)

Reviewer 3 Report

Please find the comments in the file below

Author Response

(The authors gave the same response as above.)
